# Inverting Gradient Attacks Makes Powerful Data Poisoning

**Wassim (Wes) Bouaziz**                              *wassim.s.bouaziz@gmail.com*
*Meta, FAIR & CMAP, École polytechnique*

**Nicolas Usunier**
*Work done at Meta, FAIR*

**El-Mahdi El-Mhamdi**
*CMAP, École polytechnique*

**Reviewed on OpenReview:** *https://openreview.net/forum?id=Lvy5MjyTh3*

## Abstract

Gradient attacks and data poisoning tamper with the training of machine learning algorithms to maliciously alter them and have been proven to be equivalent in convex settings. The extent of harm these attacks can produce in non-convex settings is still to be determined. Gradient attacks are practical for fewer systems than data poisoning but have been argued to be more harmful since they can be arbitrary, whereas data poisoning reduces the attacker's power to only being able to inject data points to training sets, via e.g. legitimate participation in a collaborative dataset. This raises the question whether the harm made by gradient attacks can be matched by data poisoning in non-convex settings. In this work, we provide a positive answer and show how data poisoning can mimic gradient attacks to perform an availability attack on (non-convex) neural networks. Through gradient inversion, commonly used to reconstruct data points from actual gradients, we show how reconstructing data points out of malicious gradients can be sufficient to perform a range of attacks. This allows us to show, for the first time, a worst-case availability attack on neural networks through data poisoning, degrading the model's performances to random-level through a minority (as low as 1%) of poisoned points. Code available at `https://github.com/wesbz/inverting-gradient`.

## 1 Introduction

Security in Machine Learning requires considering various attackers with a wide range of capabilities. Better understanding the capabilities of these attackers is crucial to design effective defenses and evaluations. In training time attacks, an attacker may only need to participate in the training procedure and send strategically chosen, yet legitimate-looking participation. We call *gradient attacks* the cases where the attacker can send gradients, and *data poisoning* when they send data points. The severity of attacks range from integrity attacks, where the model's reliability or consistency can be altered, to the most severe, availability attacks, where the model is plainly unusable, e.g. due to performances being too low to allow for normal operations.

So far, unless the objective function is convex Biggio et al. (2013), availability attacks have only been possible through gradient attacks Blanchard et al. (2017); El-Mhamdi et al. (2018); Baruch et al. (2019), leaving an open question, whether gradient attacks are fundamentally more powerful than data poisoning attacks outside the convex setting. **What can be the damage caused by data poisoning in a worst-case scenario, and can it match the damage of gradient attacks?** While data poisoning have been considered weaker than gradient attacks, challenging this perception is crucial to develop comprehensive security measures (Section A illustrates real-world scenarios). In this work, we answer this question by providing empirical evidence that poisoning attacks can lead to availability attacks on (non-convex) neural networks, under a worst-case threat model that allows for comparing both gradient and data poisoning attacks.

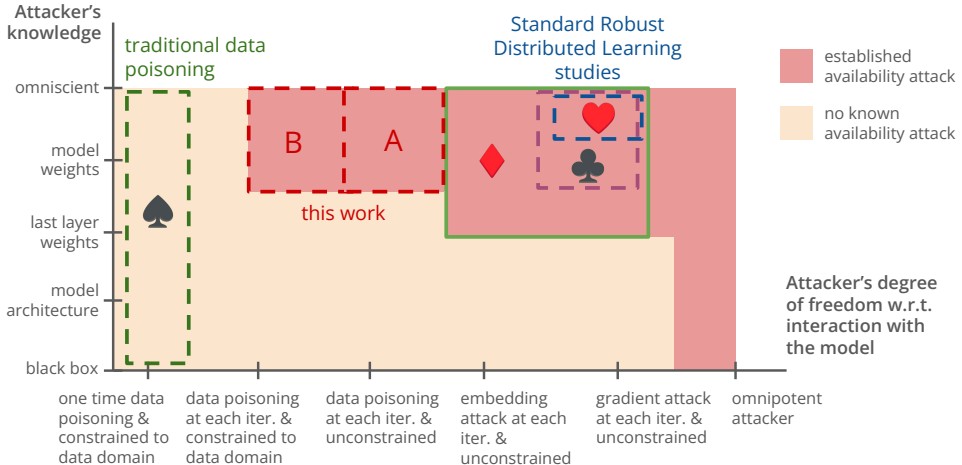

Figure 1: Territory of known availability attacks (in red) within a domain of constraints. The closer to the origin, the more constrained is the setting for the attacker to realize an availability attack. ♠: Geiping et al. (2020b); Zhao & Lao (2022); Ning et al. (2021); Huang et al. (2020), ♡: Blanchard et al. (2017); Baruch et al. (2019), ♣: El-Mhamdi et al. (2018), ◇ so far only in convex settings : Farhadkhani et al. (2022), A: Availability attacks (Result A in subsection 5.3), B: Influence of the feasible set (Result B in subsection 5.3).

An "apple-to-apple" comparison of data poisoning attacks and gradient attacks is not straightforward because the literature associates them with different threat models offering **different levels of knowledge and interventions**. Gradient attacks are only considered in Distributed Learning settings with the Byzantine Machine Learning threat model Blanchard et al. (2017); El-Mhamdi et al. (2018); Baruch et al. (2019), where legitimate workers send actual model updates and malicious workers send arbitrary updates. In contrast, data poisoning attacks are usually considered in Centralized Learning settings Biggio et al. (2013); Muñoz-González et al. (2017); Lu et al. (2022) in a threat model that only allows the attack to be crafted once and for all. In order to **remove this confounding factor**, we consider a threat model in which **both attacks** can be executed by allowing an attacker to **intervene at each iteration**, similarly to Algorithm 1 in Steinhardt et al. (2017), may it be gradients or data poisoning (Figure 1). Once this confounding factor removed, the fundamental difference between gradient and data poisoning attacks comes from the limited expressivity of data poisoning compared to the arbitrary gradient attacks.

Our approach leverages gradient inversion methods, previously used in privacy attacks in distributed learning to reconstruct training data points from actual gradients they induced. Contrary to privacy attacks, malicious gradients might not be achievable by the gradient operator if participants are expected to send legitimate-looking inputs (Figure 2). We reconstruct data points whose induced gradient can replicate as much as possible a malicious gradient, and show that they can constitute a sufficient data poisoning to achieve an availability attack.

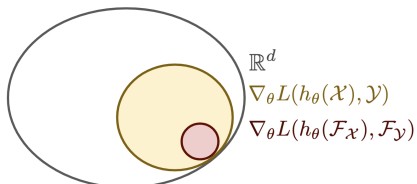

Figure 2: Images of the gradient operator on different sets. $\mathbb{R}^d$ is where an attacker can craft unrestricted gradient attacks. $\nabla_\theta L(h_\theta(\mathcal{X}), \mathcal{Y})$ is the set of possible gradients given an unrestricted data poisoning (Result B in Section 5.3), and $\nabla_\theta L(h_\theta(\mathcal{F}_\mathcal{X}), \mathcal{F}_\mathcal{Y})$ is the set of possible gradients when data poisoning is restricted to a feasible set $\mathcal{F}_\mathcal{X} \times \mathcal{F}_\mathcal{Y} \subseteq \mathcal{X} \times \mathcal{Y}$ (Result A in Section 5.3).

In our experiments, we exhibit a successful availability attack on neural network architectures, trained on an image classification task with different optimization algorithms, even when protected by a defense mechanism against gradient attacks. We show that: (1) the additional constraints under which data poisoning attacks operate, compared to gradient attacks, make them overall less effective than plain gradient attacks, and (2) the severity of data poisoning attacks covers the same range as gradient attacks, including availability attacks, even on non-convex neural networks.

Our contributions are the following:

- We leverage gradient inversion mechanism from privacy attacks to reconstruct data poisoning from existing gradient attacks and perform attacks on various settings;

- We introduce a worst-case threat model that allows for a fair comparison between gradient attacks and data poisoning attacks, by allowing the attacker to intervene at each iteration;

- We demonstrate that in our worst-case threat model, data poisoning could perform an availability attack on neural networks degrading them to random chance performances.

**Paper structure.** The rest of the paper is organised as follows. Section 2 reviews related work on the broader effort to understand various types of attacks, Section 3 describes the formal setup , Section 4 provides our solution to invert the most important gradient attacks back in the training data domain, and provides a table of correspondence between gradient attacks and data poisoning attacks, Section 5 implements our solution describing empirical outcomes, finally, Section 6 discusses our work and provides concluding remarks.

## 2 Background

**Gradient attacks** have long been studied in the standard Robust Distributed Learning literature, when an attacker can send **arbitrary**[1] gradients. Lemma 1 in Blanchard et al. (2017) shows that with stochastic gradient descent (SGD), availability attacks are possible with only a single malicious worker. The attacker's freedom allows for a variety of attacks by exploiting the geometry of honest gradients and the inevitable weaknesses El-Mhamdi et al. (2018); Baruch et al. (2019); Xie et al. (2020); Shejwalkar & Houmansadr (2021) of robust gradient aggregators when the model is high-dimensional or when the data is heterogeneous. The attacker's goal is to bring the average gradient to another direction, making a step toward their objective. There is no naive way to determine if a gradient is legitimate or not. In such settings, defending is often considered as a task of robust mean estimation. Robust aggregation of gradients is hence a possible defense mechanisms one can deploy Blanchard et al. (2017); El-Mhamdi et al. (2018); Yin et al. (2018). Still, with better defense mechanisms came better attacks Baruch et al. (2019). Even stronger, Theorem 2 in El-Mhamdi et al. (2023) shows the impossibility of robust mean estimation below a certain threshold that grows with data heterogeneity and model size. Our work relies on these gradient attacks and shows how they can be transferred to data poisoning as successful availability attacks, even in non-convex settings.

**Data poisoning** is the manipulation of training data of ML algorithms with the goal of influencing the algorithms' behavior. Several approaches exist to generate poisons: label flipping Shejwalkar et al. (2022), generative methods Muñoz-González et al. (2019); Zhao & Lao (2022), and gradient-based approaches Muñoz-González et al. (2017); Shafahi et al. (2018); Geiping et al. (2020b). The last allows to finely control the resulting gradient on the poisonous points instead of relying on another proxy. Although Shejwalkar et al. (2022); Shejwalkar & Houmansadr (2021) consider data poisoning to be of limited harm and gradient-based approach to be too computationally intensive, clean-label attacks based on gradient matching have shown to be both stealth and effective when performing a targeted integrity attack Geiping et al. (2020b). It was also demonstrated in practical settings such as state of the art image classifiers Bouaziz et al. (2025a), audio classification Bouaziz et al. (2025c), and even language modeling Bouaziz et al. (2025b). To the best of our knowledge, our work is the first to achieve an availability attack on a neural network using data poisoning.

**Availability attacks** have been demonstrated in a range of settings. As shown in Figure 1, we can compare these settings in term of attacker knowledge (from black box to omniscient) and degree of freedom (from a single constrained interaction to an omnipotent attacker). The data poisoning literature on neural networks only allows an attacker to craft a poison once to insert it in the training set, with various levels of knowledge. Geiping et al. (2020b) operate both in *black-box* setting and with sole *model architecture* knowledge. Muñoz-González et al. (2019) assume an omniscient attacker and Muñoz-González et al. (2017) add scenarios with attackers not having access to the model weights or to the training data. Farhadkhani

---

[1]Often called Byzantine attacks, in reference to the Byzantine faults model in distributed computing Lamport et al. (1982)

et al. (2022) assume an attacker who can interfere at the embedding level of the last layer of a neural network. This particular case is equivalent to attacking a logistic regression which is a convex setting for which an equivalence between data poisoning and gradient attacks holds. Previous works in data poisoning on neural networks were only able to slightly decrease the performance of the attacked algorithm and have yet to demonstrate a complete availability attack that bring a model down to chance-level Zhao & Lao (2022); Lu et al. (2022). Gradient attacks, on the other hand, have established effective ways of making a model utterly useless. Blanchard et al. (2017) show how an omniscient attacker sending unconstrained gradients at each iteration can arbitrarily change the model's weights. Baruch et al. (2019) assume an attacker with access to model's weights and a fraction of the training set (similar to our *auxiliary dataset*). El-Mhamdi et al. (2020) suppose an omniscient attacker or one that does not know the legitimate gradients. Finally, Liu et al. (2023) claim a data poisoning availability attack, but they not only require a high poisoning ratio (at least 80%) but also the degradation of performances is not as severe as in gradient attacks (Figure 3 in Liu et al. (2023)).

In this work, we extend the domain of known availability attacks and demonstrate how they are possible in settings were the attacker knows the model weights and can craft a data poisoning at each iteration in a constrained set or not, similarly to Algorithm 1 in Steinhardt et al. (2017) (Result A and B in subsection 5.3).

**Defenses**   against data poisoning have been studied through different approaches such as data sanitization Steinhardt et al. (2017), data augmentation Borgnia et al. (2021), bagging Wang et al. (2022) or pruning and fine-tuning Liu et al. (2018). However, the effectiveness of such defenses rely on strong assumptions such as the learner having access to a clean dataset, on the convexity of the loss w.r.t. the model's parameters or the learner's ability to train a very large number of models. And still, attackers could find ways to break these defenses Koh et al. (2022). Even if a theoretically sound and impenetrable defense mechanism against data poisoning might be impossible El-Mhamdi et al. (2023); Hardt et al. (2023), making the attacker's job harder by adding several imperfect yet constraining defenses in a "Swiss cheese"[2]-like model is still necessary.

**Inverting gradients**   has recently been studied in privacy attacks to reconstruct training samples based on the resulting gradient Geiping et al. (2020a); Zhao et al. (2020). Their central recovery mechanism relies on maximizing a similarity measure $Sim$ between a targeted gradient $g^{(tgt)}$ that has been computed on the data which the attacker wants to reconstruct and the gradients $G^{(rec)} = \{\nabla_{\theta_t} L(h_{\theta_t}(x), y)\}_{(x,y) \in S}$ computed on $n_{rec}$ reconstructed samples $S$ and aggregated through the AGG function:

$$\arg \max_{S \in (\mathcal{X} \times \mathcal{Y})^{n_{rec}}} Sim(g^{(tgt)}, \text{AGG}(G^{(rec)})) \tag{1}$$

Similarly, we try to recover data points that induce the closest gradient possible to a malicious gradient. The existence of a solution for (1) in the privacy attack setting is known, since actual data points have given rise to the targeted gradient $g^{(tgt)}$. More surprisingly, several works Geiping et al. (2020b); Bouaziz et al. (2025a) have shown that it is possible to construct data points whose gradients match a targeted sample's gradient. On the contrary, gradient attacks are not actually calculated from a data point hence there is no guarantee that the inversion can find an existing solution.

## 3   Framework

### 3.1   Learning Setting

We consider a classification setting where the model is trained on a dataset $D_{train} = \{(x_i, y_i)\}_{i=1}^n$ sampled from a distribution $\mathcal{D}$ over $\mathcal{X} \times \mathcal{Y}$. The learner trains a neural network $h_\theta$ parametrized by $\theta \in \mathbb{R}^d$ with an iterative optimization algorithm on the (non-convex) loss function $L$. Its goal is to achieve the lowest test loss on a heldout test set $D_{test}$ that is not necessarily sampled from the same distribution $\mathcal{D}$ as the training set. We formulate the objective as the following optimization problem:

$$\arg \min_{\theta \in \Theta} \frac{1}{n_{test}} \sum_{(x,y) \in D_{test}} L(h_\theta(x), y)$$

---

[2]a cybersecurity defense model, see `https://en.wikipedia.org/wiki/Swiss_cheese_model`

We consider that learning occurs through a set of $n_b$ *Gradient Generation Units* $\{V_i\}_{i=1}^{n_b}$ each of which reports a *message* in a set $S_t^b = \text{MESSAGE}(D_{train}, t) = \{v_{i,t}\}_{i=1}^{n_b}$ at each iteration $t$. Messages are then aggregated through an aggregator AGG and the model weights are updated using the UPDATE algorithm:

$$\theta_{t+1} = \text{UPDATE}(\theta_t, \text{AGG}, S_t^b)$$

This abstraction allows us to represent a large spectrum of learning settings, from *centralized learning* to fully *distributed learning* and settings in between (such as *federated learning*).

- In the common *centralized setting*, when training a neural network, the MESSAGE operator returns a batch of data points sampled from the training set $S_t^b = \{(x_{i,t}, y_{i,t})\}_{i=1}^{n_b}$, the aggregator AGG is the AVERAGE of their gradients, computed by the learner and the update algorithm is SGD or ADAM.

- In the *federated learning setting*, the MESSAGE operator returns a batch of gradients (or equivalently model updates) that each worker computed separately, the default aggregator is the AVERAGE of the messages and the update algorithm is FEDERATEDAVERAGING McMahan et al. (2017) or LOCALSGD Stich (2018).

This allows us to **study a common framework for both gradient attacks and data poisoning attacks**, normally operating on different settings, characterized by different MESSAGE operators. The choice for the MESSAGE operator boils down to entrusting the calculation of gradients to the workers or not. This, in turn, gives different degrees of freedom for a malicious *Gradient Generation Unit* to influence the training: in the scope of this paper, respectively gradient attacks or data poisoning[3].

In our experiments, we consider an image classification task on the CIFAR10 dataset on which the attacker can tamper with messages (data point or gradient depending on the learning setting). We use the SGD and ADAM update algorithms as well as AVERAGE and MULTIKRUM Blanchard et al. (2017) aggregators. MULTIKRUM$_{f<0.5}$ is a robust aggregator which can withstand a ratio of malicious elements up to $f$ and is defined for a set of vectors $\{v_i\}_{i=1}^n$ as the average of the $n(1-f) - 2$ vectors minimizing the score function $s(i) = \sum_{i \to j} \|v_i - v_j\|^2$, with $i \to j$ the indices of the $n(1-f) - 2$ closest vectors to $v_i$.

The selection of the update algorithm and aggregator is crucial, as both are regarded as defense mechanisms in the Robust Distributed Learning literature El-Mhamdi et al. (2020).

## 3.2 Threat model

In order to study the expressivity of data poisoning in non-convex settings, we consider **a worst-case threat model** that allows for a fair comparison with gradient attacks. Here, the attacker:

- has knowledge of the weights of the model $\theta_t$, the update algorithm UPDATE, the aggregator AGG function and the message operator MESSAGE as in, Farhadkhani et al. (2022); Blanchard et al. (2017); Yin et al. (2018); Shafahi et al. (2018);

- does not have access to the batch $S_t^b$, unlike the stronger standard assumption of omniscient attacker in the robust distributed learning literature Blanchard et al. (2017); El-Mhamdi et al. (2020); Chen et al. (2017); Yin et al. (2018);

- has access to an auxiliary dataset $D_a \sim \mathcal{D}$ that is a surrogate to the unobservable training set. This is a standard assumption when not assuming omniscient attackers, as in El-Mhamdi et al. (2018); Farhadkhani et al. (2022); El-Mhamdi et al. (2020);

- has the control over a ratio $\alpha$ of *Gradient Generation Units* and the ability to append a set of arbitrarily crafted poisoned messages (gradients or data points) $S_t^p = \{v_{i,t}^p\}_{i=1}^{n_p}$ to the clean batch $S_t^b$ at each iteration $t$ up to a level $\alpha$ of contamination (i.e. $|S_t^p|/|S_t^{b\cup p}| = \alpha$), as in Blanchard et al.

---

[3]The notion of *Gradient Generation Units* can be used as an abstraction for any source of input that is itself a gradient, a model update, or that can be later transformed to a gradient, such as a training data point, a machine in distributed learning or a user account in a social media platform, depending on the level of granularity that is considered.

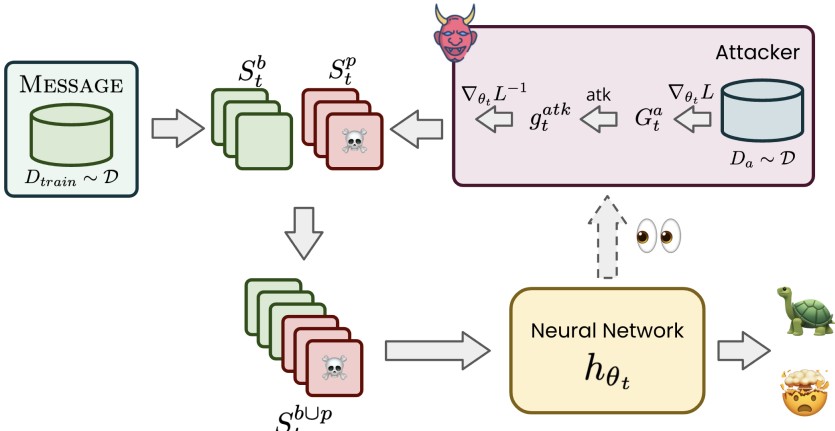

Figure 3: Threat model. The attacker has access to $\theta$ but does not have access to the batch $S_t^b$ and uses an auxiliary dataset $D_a$ to craft $S_t^p$ the set of poisoned messages. Both the batch and the poisons set are gathered into $S_t^{b \cup p}$. The attacker's goal is either to slow down the training or attack the model's availability.

(2017); El-Mhamdi et al. (2018); Yin et al. (2018) for gradients attacks, in Steinhardt et al. (2017) for data poisoning or in Farhadkhani et al. (2022) for both gradients attacks and data poisoning;

- constrains itself[4] to only crafting poison in a feasible domain $S_t^p \in \mathcal{F}^{n_p}$, depending on the task and data structure. For instance, for an image classification task of $H \times W$ RGB pixels encoded on 3 bytes and labels between 1 and $C$, we set $\mathcal{F}$ to be the set of possible such sized images and labels: $\mathcal{F} = [0..255]^{H \times W \times 3} \times [1..C]$ as in Farhadkhani et al. (2022), but without restricting ourselves to convex settings.

The attacker's goal is to perform an **availability attack:** degrading the learner's performances as much as possible. We also consider **slow down attacks**, in which the attacker tries to stall the learning procedure as much as possible. This attack is important, given the cost for training large models and the financial repercussion of slowing it down.

This threat model is inspired from gradient attacks and assumes much more communication between the attacker and the model than traditional data poisoning threat models. The latter leads to poor availability attacks performance. We hypothesized that once we allowed for the same amount of interactions between gradient attacks and data poisoning, the two methods would display far less discrepancy of destructive power than what is currently thought.

## 4 Method

At each iteration $t$, the attacker, controlling $p$ *Gradient Generation Units*, computes an auxiliary batch $S_t^a = \text{MESSAGE}(D_a, t)$ and constructs $S_t^p$ such that $\text{UPDATE}(\theta_t, \text{AGG}, S_t^a \cup S_t^p)$ gives poorer performance on $D_a$ than $\text{UPDATE}(\theta_t, \text{AGG}, S_t^a)$. This should, in turn, also decrease the model's performances on $D_{train}$ under the assumption it follows the same distribution as $D^a$.

### 4.1 Gradient attacks

When the MESSAGE operator outputs gradients, a malicious *Gradient Generation Units* can participate in $S_t^b$ with a gradient attack in $\mathbb{R}^d$. At each iteration $t$, the attacker uses the auxiliary dataset $D_a$ as a surrogate

---

[4]Result A in subsection 5.3 is our strongest result and is obtained under this constraint on the attacker. Result B is 5.3 is obtained without this constraint and serves to further understand the expressivity of data poisoning from inverted gradient attacks.

for the batch $S_t^b$. Let $S_t^a = \{g_{t,i}^a\}_{i=1}^{n_a} = \{\nabla_{\theta_t} L(h_{\theta_t}(x), y)\}_{(x,y) \in D_a}$ be the set of per-sample auxiliary gradients and $g_t^a = \frac{1}{n_a} \sum_{g \in S_t^a} g$ the averaged auxiliary gradient. Similarly, $g_t^b = \frac{1}{n_b} \sum_{g \in S_t^b} g$ denotes the averaged clean batch gradient. The attacker constructs a set of $n_p$ attacking gradients $G_t^{atk}$ based on $S_t^a$ and sends them as message $S_t^p$. We denote $S_t^{b \cup p} = S_t^b \cup S_t^p$ the set of poisoned batch of gradients, $S_t^{a \cup p} = S_t^a \cup S_t^p$ the set of poisoned auxiliary gradients and $g_t^{b \cup p}$ and $g_t^{a \cup p}$ their respective averages. We consider several gradient attacks to perform an availability attack.

**Gradient Ascent** (GA) Blanchard et al. (2017): the attacker sends a gradient such that the averaged poisoned gradient is anti-collinear with the mean clean gradients. This provokes a gradient ascent step with the SGD update rule, AVERAGE, and $\lambda > 0$:

$$\mathbb{E}_{S_t^b}[\theta_{t+1}] = \theta_t - \eta \mathbb{E}_{S_t^b}[g_t^{b \cup p}]$$
$$= \theta_t + \eta \lambda \mathbb{E}_{S_t^b}[g_t^b]$$

**Orthogonal Gradient** (OG): the attacker sends a gradient such that the averaged poisoned gradient is orthogonal to the mean clean gradient. This should stall the training.

**Little is Enough** (LIE) Baruch et al. (2019); Shejwalkar & Houmansadr (2021): the attacker sends the mean clean gradient deviated by a strategically chosen amount times the coordinate-wise standard deviation of the clean gradients, with $\sigma[j] = \sqrt{Var(\{g[j]\}_{g \in S_t^a})}$.

$$g_t^{atk} = g_t^a - z^{max}\sigma \tag{2}$$
$$\text{where} \qquad z^{max} \in \arg\max_{z \in \mathbb{R}_+^*} \left\| A_t^{a \cup p} - A_t^a \right\|,$$
$$\text{and} \qquad A_t^{a \cup p} = \text{AGG}(S_t^a \cup \{g_t^a - z\sigma\}_{i=1}^{n_p})$$
$$A_t^a = \text{AGG}(S_t^a)$$

This attack has been shown to be effective against MULTIKRUM aggregator. Note that contrary to Baruch et al. (2019) and similarly to Shejwalkar & Houmansadr (2021), we use an adaptive approach and choose $z^{max}$ to maximize the divergence of the poisoned gradients on the aggregation $A_t^{a \cup p}$.

## 4.2 Data poisoning

When the MESSAGE operator outputs data samples, malicious *Gradient Generation Units* are expected to participate via similarly structured messages, i.e. data poisoning. In our experiments, on CIFAR10 and its $32 \times 32$ RGB images, it leaves room for an attacker to participate in the training process via messages crafted in $[0, 255]^{32 \times 32 \times 3}$. This gives only $32 \times 32 \times 3 = 3,072$ degrees of freedom, far less than commonly used deep learning models' number of parameter, hinting that data poisoning should be less expressive than gradient attacks. Models' non-linearities further constrain the dimension of the image space of the gradient operator (as illustrated in Figure 2). Data poisoning is allegedly harder as it constitutes a far more constrained problem than gradient attacks (in which the attacker can send an arbitrary gradient from $\mathbb{R}^d$).

At each iteration $t$, the attacker computes a given gradient attack $g_t^{atk}$ in the same manner as above and inverts the gradient operator to compute an associated set of data points $S_t^p = \{(x_{i,t}^p, y_{i,t}^p)\}_{i=1}^{n_p}$ such that $\frac{1}{n_p} \sum_{i=1}^{n_p} \nabla_{\theta_t} L(h_{\theta_t}(x_{i,t}^p), y_{i,t}^p) = g_t^{atk}$. That set of data points is then sent as message $S_t^p$ and appended to $S_t^b$, like in the gradient attack case.
Similarly to what is done in privacy attacks, we optimize a similarity measure between the gradient attack and the gradients calculated on $S_t^p$. Contrary to privacy attacks where the existence of a solution is known[5], there is no guarantee that the inversion can find an existing solution, let alone an effective data poison.

We can formulate our data poisoning as an optimization problem where the attacker aims to minimize a poisoning function $f_p$. $f_p$ characterizes the dissimilarity of the data poison gradients $G_t^p = \{\nabla_{\theta_t} L(h_{\theta_t}(x), y)\}_{(x,y) \in S_t^p}$

---

[5]In privacy attacks, the gradient being inverted was produced by the data point that the attacker is trying to retrieve.

with a gradient attack $g_t^{atk}$ calculated from the auxiliary dataset gradients $G_t^a = \{\nabla_{\theta_t} L(h_{\theta_t}(x), y)\}_{(x,y) \in S_t^a}$ (since the attacker cannot have access to the batch $S_t^b$) over a feasible domain $\mathcal{F}$:

$$S_t^p \in \arg \min_{S \in \mathcal{F}^{n_p}} f_p(h_{\theta_t}, G_t^a, S) \tag{3}$$

Since characterizing the image space of the gradient operator on the loss function $L$ is a difficult task, we cannot know beforehand if a vector can have an antecedent data point through the gradient operator. Together with constraints on feasibility, finding data poisons which exactly reproduce the gradient attack might be impossible. Since $\mathcal{I}_{\nabla \mathcal{F}} = \nabla_{\theta_t} L(h_{\theta_t}(\mathcal{F}_{\mathcal{X}}), \mathcal{F}_{\mathcal{Y}}) \subseteq \nabla_{\theta_t} L(h_{\theta_t}(\mathcal{X}), \mathcal{Y}) \subseteq \mathbb{R}^d$, the set $\mathcal{I}_{\nabla \mathcal{F}}$ might not cover all the possible candidates for an effective gradient attack, if not any (Figure 2). Thus, achieving an effective data poisoning that performs similarly to a gradient attack is non trivial.

We show in our experiments that an attacker can still produce data poisons which have significant impact on the training procedure. Poisons are iteratively updated to minimize the poisoning objective $f_p$ using the `Adam` optimizer and are projected on the feasible set $\mathcal{F}$ at each iteration of the poisoning optimization by clipping. Table 1 details the formulas used to determine the gradient attacks and their equivalent poisoning functions in the data poisoning case.

Table 1: Gradient attacks formula and their equivalent data poisoning objective functions to be optimized following Equation 3. $g^p, g^a, g^{a \cup p}$ are respectively the averaged gradients computed on the data poisons, on the auxiliary dataset, and the weighted average between them. cos is the cosine similarity. $z^{max}$ and $\sigma$ are defined as in eq. 2.

|  | **Gradient attack** $g^p \in \mathbb{R}^d$ s.t. | **Our data poisoning attack** $f_p$ |
|---|---|---|
| Gradient Ascent | $\cos(g^{a \cup p}, g^a) = -1$ | $\cos(g^{a \cup p}, g^a)$ |
| Orthogonal Gradient | $\langle g^{a \cup p}, g^a \rangle = 0$ | $\|\cos(g^{a \cup p}, g^a)\|^2$ |
| Little is Enough | $g^p = g^a - z^{max} \times \sigma$ | $\|g^p - g^a + z^{max}\sigma\|^2$ |

## 5 Experiments

### 5.1 Preliminary experiment

To give a better intuition of the capabilities of data poisoning, we present a preliminary experiment on the `XOR` operator classification task. Sampling a point $x$ in $[0,1]^2$, its label is $y = \mathbb{1}\{x[0] > 0.5\} \oplus \mathbb{1}\{x[1] > 0.5\}$ with $\mathbb{1}$ the indicator function and $\oplus$ the `XOR` operator. We generate a set of possible poisons by regularly sampling on the $[0,1]^2$ grid and labeling them with `XOR` then **flipping** their labels. After training a multilayer perceptron with Gradient Descent (i.e. full batch) on 1000 data points, we perform a single step of gradient descent on the data poisoned with one of the possible poisons, repeated as to reach a contamination level $\alpha$. The resulting test accuracies obtained for all poisons are illustrated in Figure 4.

While a single step at $\alpha = 0.005$ is not enough to reduce the model accuracy down to random-chance, several steps of poisoned update will do. A 10% level of contamination is already enough to bring the model close to the level of a random guess, which suggests that a single step on poisoned data is enough to obtain an availability attack on neural network for a task such `XOR`. This proof of concept could however be argued to require a high contamination ratio. We exhibit experimental evidences that lower contamination ratio can actually be effective on, e.g. an image classification setting.

### 5.2 Experimental setup

**Model & dataset**  We demonstrate our poisoning procedure on a custom convolutional neural network (described in Table 5 in Appendix B) and on Vision Transformers models (`ViT-tiny` models with patch size 8) trained for 50 epochs on the CIFAR10 dataset partitioned in training, validation, and auxiliary datasets. We use different optimization algorithms and aggregation rules to train the models: SGD & AVERAGE,

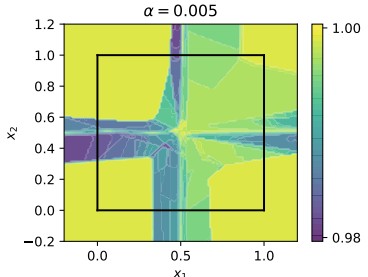 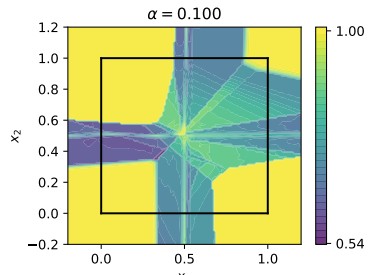 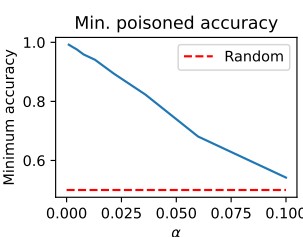

Figure 4: Landscape of accuracies after 1 poisoned step with the respective poison with two levels of contamination $\alpha = 0.005$ (**left**) and $\alpha = 0.1$ (**center**). The black squares represent the border of the feasible domain $\mathcal{F}$. **Right:** The minimum accuracy in the landscape for different levels of contamination $\alpha$.

ADAM & AVERAGE, SGD & MULTIKRUM (with different levels of data truncation $f \in \{0.1, 0.2, 0.4\}$). Since MULTIKRUM is used as a defense mechanism, we expect it to be robust to the attacks operating within its working assumptions. However, the Little is Enough attack was specifically designed to circumvent these assumptions when using gradient attacks. We thus expect Little is Enough to bypass MULTIKRUM in the gradient attacks situation.

**Baseline**  In every setting, the learning rate is fixed to the value were the learner achieves the best performances without any poisoning to set a baseline for the performances of the model. We then measure the decay in performances caused by an attack. Each setting of this experiment is run 4 times for better statistical significance. Each run has a different model and poisons initialization, and dataset split. Full results can be seen in the Appendix.

**Attacks**  In every setting, we perform either one of the considered attacks either via a gradient attack, or a data poisoning attack as specified in section 4.2. The $n_p$ crafted poisons are added to the batch of size $n_b$ at each iteration so that $\frac{n_p}{n_b + n_p} = \alpha \in [0.01, 0.48]$. Since the gradients induced by the poisoned data mimic the malicious gradients, we expect our data poisoning is at best as good as the associated gradient attack. Gradient attacks should thus be a topline for our data poisoning attack.

## 5.3   Results

### 5.3.1   Gradient attacks

Table 2 shows that gradient attacks do perform an availability attack, bringing the models' performances down to random-level. We also notice that the MULTIKRUM aggregation rule does act as a defense mechanism, for levels of contamination lower than its tolerance parameter $f$. However, as expected, this defense is well circumvented by the Little is Enough attack.

### 5.3.2   Data poisoning

After performing the equivalent data poisoning attacks, the observed effects range from a slowdown of the training procedure to its complete halt and up to degrading the performances down to random levels. Figure 5a compares the different attacks with a random data poisoning (uniformly sampled in $[0, 1]$) at $\alpha = 0.01$ with its baseline counterpart. While the Gradient Ascent and Orthogonal attacks behave similarly and at most only slightly slow down the training at most, Little is Enough attack strongly degrades performances, even at such low levels of contamination. Figure 5b shows for the Gradient Ascent and Little is Enough attacks that the higher the level of contamination, the stronger the effect: validation accuracy increases slower or plummets earlier. Therefore, our main result is the following.

**Availability attacks (Result A).**  To fairly compare each setting under that attack, each model has been evaluated on the test set with the weights achieving the best validation accuracy. Figure 6 shows that for

Table 2: Best validation accuracy under different attacks with different update rules, different aggregation functions and different levels of contamination $\alpha$. A high validation accuracy (colored in apricot) indicates a failed attack. A low validation accuracy (colored in pale green) indicates a successful attack.

| Update; Agg | Attack | $\alpha$ | | | | | | |
|---|---|---|---|---|---|---|---|---|
| | | 0.01 | 0.05 | 0.10 | 0.20 | 0.30 | 0.40 | 0.48 |
| Adam; Average | GA | 10.0 | 9.8 | 10.0 | 10.0 | 10.0 | 10.1 | 10.0 |
| | OG | 9.9 | 9.9 | 10.1 | 10.1 | 9.8 | 10.0 | 10.1 |
| | LIE | 10.2 | 10.3 | 10.1 | 10.2 | 10.1 | 10.1 | 10.0 |
| SGD; Average | GA | 10.1 | 10.1 | 10.1 | 9.9 | 10.0 | 10.0 | 10.0 |
| | OG | 10.2 | 10.2 | 10.0 | 9.9 | 10.1 | 10.1 | 10.2 |
| | LIE | 10.2 | 9.9 | 10.2 | 9.9 | 9.9 | 10.0 | 10.0 |
| SGD; MultiKrum$_{f=0.1}$ | GA | 65.1 | 63.2 | 33.7 | 10.1 | 10.1 | 10.2 | 10.2 |
| | OG | 65.1 | 65.3 | 65.9 | 10.0 | 10.1 | 10.2 | 9.9 |
| | LIE | 41.7 | 15.0 | 10.4 | 9.9 | 10.0 | 9.9 | 10.1 |

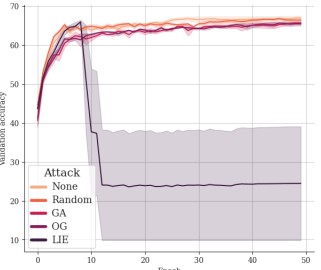

(a) Comparison of different attacks at $\alpha = 0.01$.

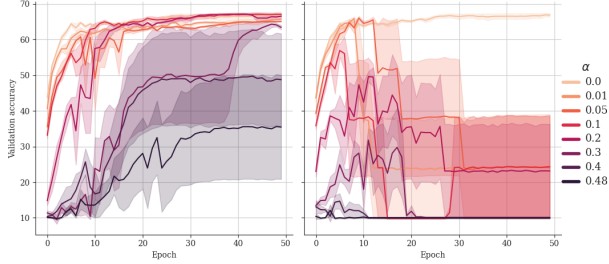

(b) Comparison of different levels of contamination for the Gradient Ascent & Little is Enough attacks.

Figure 5: Validation accuracies during training in the SGD & Average setting under different attacks and different level $\alpha$ of contamination. Error bars represent the standard error.

the SGD & Average learner under the Little is Enough attack, $\boldsymbol{\alpha = 0.01}$ **is enough to significantly degrade the model's performance**. Among the 4 runs, 3 ended up diverging while every poison was **in the feasible set** $\mathcal{F}$. Figure 9 in the Appendix shows 50 of the first 500 poisons crafted by the attacker in one of the diverging run. Similarly, in the SGD & MultiKrum$_{f=0.1}$ setting, the Little is Enough attack with contamination level $\boldsymbol{\alpha = 0.05}$ **finds data poisons that circumvent the robust aggregation rule** and drastically reduce the model's performance.

Table 3: Epoch of best validation accuracy (validation accuracies in parenthesis) for the CNN model with Adam optimizer and Average under different data poisoning attacks. Higher levels of contamination induce slower training and lower performances.

| $\alpha$ | GA | OG | LIE |
|---|---|---|---|
| **0** | 4 (67.1) | 4 (67.1) | 4 (67.1) |
| **0.01** | 5 (66.4) | 5 (66.4) | 6 (66.3) |
| **0.05** | 7 (66.5) | 7 (65.3) | 7 (66.3) |
| **0.1** | 19 (63.6) | 14 (63.9) | 10 (65.7) |
| **0.2** | 34 (61.2) | 17 (62.5) | 22 (62.1) |

Table 4: Poison selection rates in the SGD & MULTIKRUM$_{f=0.1}$ averaged over all runs and all epochs. Each row corresponds to a different attack. Standard deviations in parenthesis.

| | | | $\alpha$ | |
| --- | --- | --- | --- | --- |
| | **0.01** | **0.05** | **0.1** | **0.2** |
| **GA** | 0.0 | $2 \times 10^{-4}$ | $2 \times 10^{-3}$ | **0.91** |
| | (0.0) | $(4 \times 10^{-4})$ | $(2 \times 10^{-3})$ | (0.08) |
| **OG** | 0.0 | $3 \times 10^{-4}$ | 0.18 | 0.85 |
| | (0.0) | $(1 \times 10^{-3})$ | (0.1) | (0.05) |
| **LIE** | **0.92** | **0.91** | **0.97** | 0.87 |
| | (0.07) | (0.13) | (0.02) | (0.06) |

**Comparison of update rules.** Figure 6 shows that our data poisoning procedure failed at performing an availability attack against the ADAM update rule. Because ADAM normalizes the aggregated gradients, the training cannot diverge as abruptly as with SGD. However, the slow down attack can still be observed (e.g. in Figure 8 in Appendix B), meaning that the data poisoning procedure could find a solution that perturbs the total gradient enough to slow down convergence, but not enough to completely halt it. Table 3 shows that higher levels of contamination lead to a lower best validation accuracy but most importantly to a higher number of epochs (hence a longer time) before reaching it.

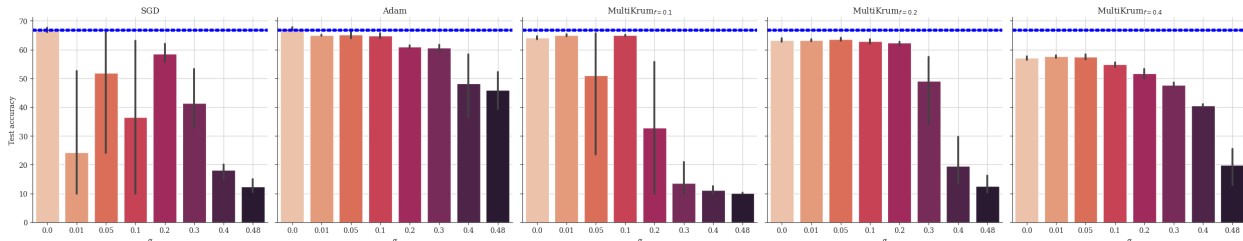

Figure 6: Test accuracy of the CNN model which achieved the best validation accuracy under the Little is Enough attack. Each column represents a different setting of update function and aggregation rule. The blue line is the test accuracy obtained with (SGD, AVERAGE) setting and no attack. Error bars are the standard errors.

**Data poisoning against a robust aggregation rule.** Table 4 shows that with the MULTIKRUM$_{f=0.1}$ aggregation rule, the attacker still manages to have some of its messages not filtered out. For $\alpha > f$, the aggregation rule does not play a defensive role anymore as per its functioning conditions. On the other hand, for $\alpha$ below this point, the Little is Enough attack displays significantly higher selection rates than the other attacks. This means that the attacker manages to produce data poisons whose gradients deceive MULTIKRUM$_{f=0.1}$, similarly to the gradient version of the attack which is particularly designed for this purpose. However Figure 6 shows that a higher selection rate does not necessarily mean success of attack. Even if the attacker sometimes manages to successfully attack the model, MULTIKRUM overall enhances the robustness of the model while slightly degrading its performances.

**Influence of the feasible set (Result B).** As the feasible set $\mathcal{F}_{\mathcal{X}}$ changes, the attacker will only be allowed to converge (in case of convergence) towards different poisons. We compare three increasingly restrictive feasible sets to determine their influence on the success of the attack:

- Constraint-free set: the feasible set is simply the input domain $\mathcal{F}_{\mathcal{X}\text{free}} = \mathbb{R}^{H \times W \times 3}$;
- Image-encoding set: the feasible set ensures that the poisons respect the same encoding as the clean data $\mathcal{F}_{\mathcal{X}\text{img}} = [0..255]^{H \times W \times 3}$; and
- Neighborhood set: this is a subset of the previous one and is composed of all the images that are at a L1 norm of at most $\epsilon = \frac{32}{255}$ of an actual image $\mathcal{F}_{\mathcal{X}\text{nei}} = \{x \in \mathcal{F}_{\mathcal{X}\text{img}}/\exists x_a \in D_a s.t. \|x - x_a\|_1 \leq \epsilon\}$.

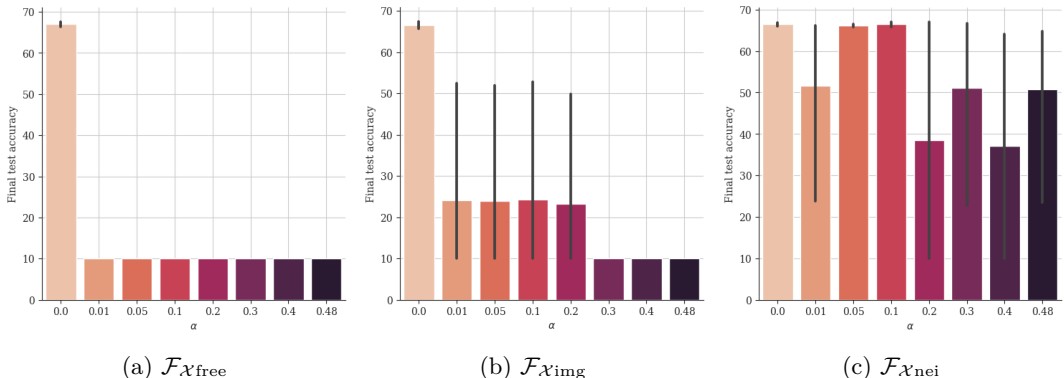

Figure 7: Final test accuracies for the SGD; AVERAGE setting under the Little is Enough attack for different feasible sets. Error bars are the standard errors.

Figure 7 shows that the more constrained the feasible set, the less effective the resulting attack. It is important to note, however, that early stopping helps in limiting the effects of the attack (Figure 10, Appendix B).

**Neural Network architecture.** We perform the same attacks on the same learning pipelines replacing the CNN with a Visual Transformer (ViT, Dosovitskiy et al. (2020)) tiny with patch size 8. Since ViTs are predominantly trained using ADAM-like approaches, we report the efficiency of our attacks using it. Figure 11 in the Appendix shows that ViTs are also vulnerable to our data poisoning, although with less success and on higher levels of contamination.

## 6 Discussion

Our work shows that even in training settings involving deep neural networks, which are not restricted to convex cases as in Farhadkhani et al. (2022), inverting malicious gradients can result in an effective data poisoning, achieving an availability attack, and mimicking the harm of gradient attacks. To compare both type of attacks, our threat model uses assumptions that differ from the common threat model in data poisoning, making it an impractical threat model for now. These assumptions should be further explored in future work on at least four different fronts. **(1) The role of the auxiliary dataset**. Further experiments should consider an auxiliary dataset with a distribution different than the training set or no auxiliary dataset at all. **(2) Accessing the trained model's weights**. Performing an attack when the attacker estimates the victim's model with a surrogate model would make the threat model more practical. **(3) The role of the feasible set**. Clean label poisoning (Geiping et al., 2020b; Shafahi et al., 2018) shows that it is possible to design data poisons which deceive human annotators by resembling legitimate images. Future work should consider exploring more constraining feasible sets for the attacker to reach such level of stealthiness. **(4) Number of interventions**. In our threat model, similarly to Farhadkhani et al. (2022) the attacker is allowed to craft and inject malicious data at each iteration, which limits the practicality of our threat model, future work should explore data poisoning availability attacks limiting also the **number of times the attacker can craft its poisons** (the common data poisoning threat model only allows a single intervention).

## 7 Conclusion

Our work asks whether or not gradient attacks were fundamentally more effective than data poisoning. Our experiments yield a nuanced response. On one hand, inverting malicious gradients sometimes results in a devastating data poisoning, and our results are the first to show the feasibility of a total availability attack on neural network via data poisoning. On the other hand, the success rate of our data poisoning attacks is lower than for their gradient attack counterparts, but the mere possibility of mimicking gradient attacks with feasible data poisoning should motivate further research in defense mechanisms for attacks that evade today's state of the art defense in robust machine learning.

## 8 Broader Impact Statement

This paper presents work whose goal is to advance the field of Robust Machine Learning. There are many potential societal consequences of our work, all of which fall under the usual considerations to take into account when considering tools that can facilitate data poisoning or make it more potent. In particular, our technique of inverting gradients aims at showing that availability attacks – which were believed to be specific to gradient attacks except for convex settings – are doable with data poisoning alone, and in small amount.

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

# Appendix

## A   Discussion on realistic values of $\alpha$

Machine Learning practitioners process data scraped online Luccioni & Viviano (2021); Schuhmann et al. (2022) or sent by users and trust their resulting models Hoang et al. (2021). Since data poisoning has proven to be practical in real case scenario Carlini et al. (2023), we should treat models trained on scrapped or collaborative datasets with as much precaution as untrustworthy data. A few redditors unpurposefully obtained a dedicated token in GPT-3's tokenizer by artificially inflating their online presence on the platform by massively posting over 160k posts on the "r/counting" subreddit on which people simply count [6]. Even worse, these data can potentially be sent by malevolent agents who can thus influence the models by legitimately participating to these data sources.

Estimating a realistic value for $\alpha$ is difficult since the point for attackers is to not be detected. Since bots represent a non-negligible part of online users (between 5% [7] and 15% Varol et al. (2017) on Twitter), we could expect the ratio of stealthy malevolent agents to be somewhat similar (if not higher). "Armies" as large as 2 million full-time individuals Charon & Vilmer (2021) could be conducing campaigns on social media (each individual manually steering tens of social media accounts to escape automated-activity detection). Collaborative datasets like Wikipedia Carlini et al. (2023) could already be stealthily poisoned. As such, we should not only presume that data we train on might have been poisoned, we should also consider the contamination ratio to be much higher than a fraction of a percent. In this work, we considered a wide range up to the extreme case of $\alpha = 0.48$

We chose to solve the poisoning optimization problem with gradient-based approaches which are computationally intense, limiting us in experimenting with larger models and datasets. Stronger gradient attacks that only require to steer the model in a direction only slightly dissimilar from honest gradients should also be explored to increase the chances for a stealth data poison to exist and to improve the attack success rate. Stronger or less computationally expensive data poisoning approaches (that may not rely on gradient attacks) can be experimented to enhance the attack success rate. While the tested defense mechanisms for the gradient case appear to generalize and defend against data poisoning, it has been shown that the latter can bypass certain defense mechanisms Koh et al. (2022) and that robust mean estimation only works up to a certain point El-Mhamdi et al. (2023). This leaves room for potentially devastating data poisoning that can bypass defenses while mimicking an unstoppable gradient attack.

## B   Complementary Figures and Tables

Table 5: Architecture of the 1.6M parameters convolutional neural network used for our experiments.

| Layer | # of channels | kernel | stride |
|---|---|---|---|
| Conv2d | 32 | $5 \times 5$ | 2 |
| ReLU | | | |
| Conv2d | 64 | $5 \times 5$ | 2 |
| ReLU | | | |
| Linear | 512 | | |
| ReLU | | | |
| Linear | 64 | | |
| ReLU | | | |
| Linear | 10 | | |

---

[6]SolidGoldMagikarp (plus, prompt generation)
[7]https://twitter.com/paraga/status/1526237585441296385

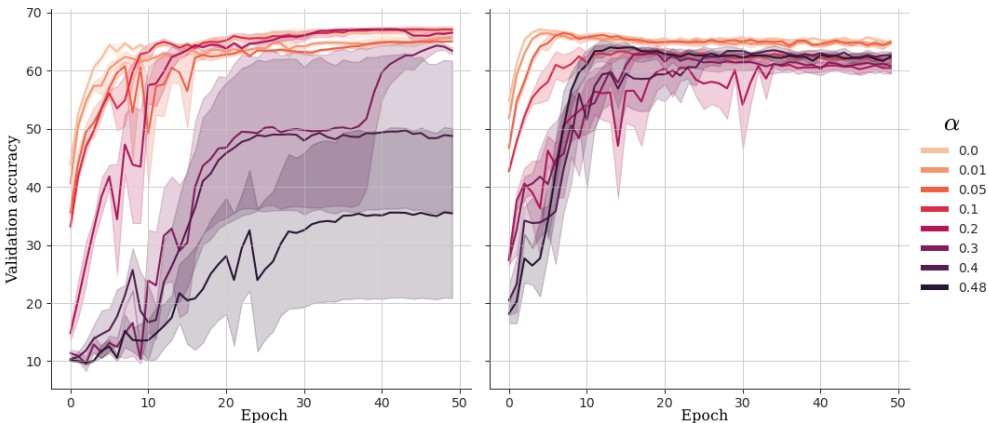

Figure 8: Validation accuracies of the CNN during training with the SGD and ADAM update rule with the AVERAGE aggregation function under the Gradient Ascent attack. This data poisoning manages to slow down the training but not degrade the model's performance to random levels. Error bars represent the standard error.

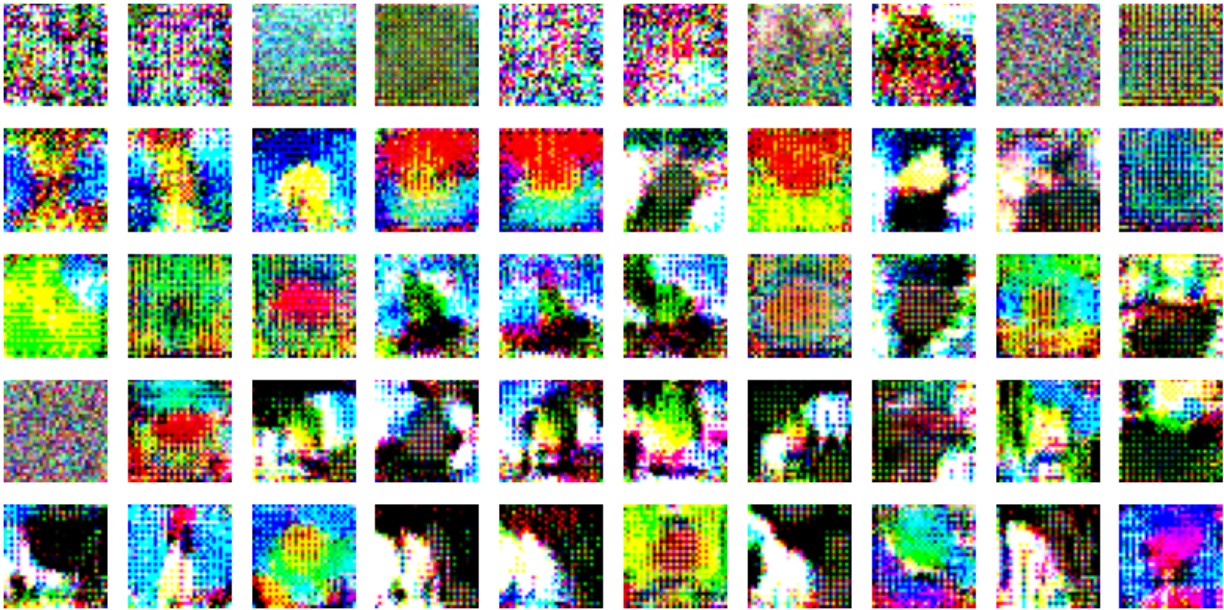

Figure 9: 50 of the first 500 poisons crafted in the (SGD & AVERAGE, Little is Enough, $\alpha = 0.01$) setting.

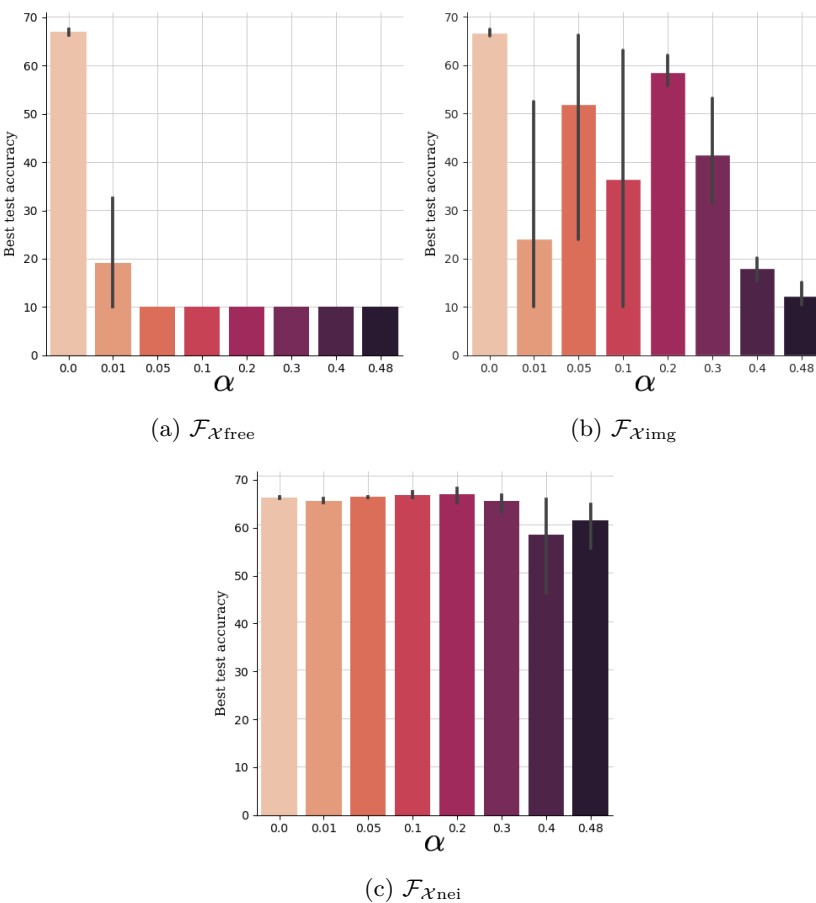

(a) $\mathcal{F}_{\mathcal{X}\text{free}}$   (b) $\mathcal{F}_{\mathcal{X}\text{img}}$

(c) $\mathcal{F}_{\mathcal{X}\text{nei}}$

Figure 10: Best test accuracies for the SGD; AVERAGE setting under the Little is Enough attack for different feasible sets.

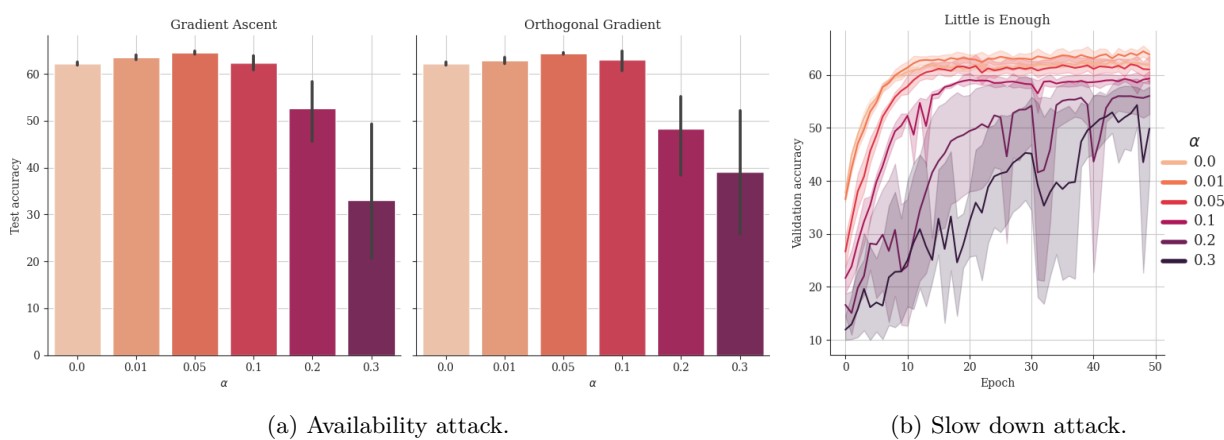

(a) Availability attack.   (b) Slow down attack.

Figure 11: Visual Transformer (ViT) tiny with patch size 8 under different attacks. Little is Enough performs a slow down attack whereas Gradient Ascent and Orthogonal Gradient are able to perform an availability attack for high enough contamination levels $\alpha$.

