# OpenReview forum: "Inverting Gradient Attacks Makes Powerful Data Poisoning"
_TMLR — Accepted by TMLR_

### Review · Reviewer_Xoxo · 2025-08-25

**Summary Of Contributions:**

Summary:
This paper proposes a novel and effective method for data poisoning attacks on non-convex neural networks. The core contribution is the idea of using gradient inversion (a technique typically associated with privacy attacks) to craft powerful data poisons. The authors demonstrate that by first designing a malicious gradient (as in a standard gradient attack) and then inverting it to find a corresponding data point, they can create a poisoning attack that effectively mimics the devastating impact of a direct gradient attack.

This approach allows them to perform a worst-case availability attack, successfully degrading a model's performance to the level of random guessing with as little as 1% poisoned data. The paper establishes a threat model that enables a direct and fair comparison between the capabilities of data poisoning and gradient attacks by allowing the attacker to intervene at each training iteration.

Strengths:

The connection drawn between gradient inversion, gradient attacks, and data poisoning is a creative and significant contribution to the field of adversarial machine learning.

The authors provide comprehensive experiments on CIFAR10 with both CNN and Vision Transformer architectures, showing the attack's effectiveness against different optimizers (SGD, ADAM) and a common defense mechanism (MultiKrum).

The paper is well-structured and clearly articulates the problem, the proposed method, and the experimental results.

Weaknesses:

The assumption that an attacker can inject poisons at every training iteration and has full knowledge of the model's weights is a significant one and may not be practical in all real world scenarios.

While testing against MultiKrum is valuable, the evaluation would be more robust if it included a broader range of modern data poisoning defenses.

**Audience:**

Yes

**Audience Explanation:**

The findings of this paper would be of significant interest to the TMLR audience, particularly researchers and practitioners in machine learning security, robustness, and privacy. The paper addresses a fundamental question in adversarial ML that is, are gradient attacks fundamentally more powerful than data poisoning? By showing an equivalence in destructive capability (under their threat model), the authors challenge a common assumption and open up new avenues for both attack and defense research. The novel use of gradient inversion for this purpose is a clever technique that will likely inspire further work in this area.

**Claims And Evidence:**

Yes

**Claims Explanation:**

The claims are well-supported by a thorough set of experiments. The authors convincingly demonstrate that their method can achieve an availability attack, a feat that has been difficult to accomplish with previous data poisoning techniques in non-convex settings.

The results consistently show that the attack can reduce model accuracy to random-chance levels (around 10% on CIFAR10), which is a clear indicator of a successful availability attack.

The attack is shown to be effective even with a very small fraction of poisoned data (e.g., 1-5%), highlighting its potency.

The attack is shown to be effective against both SGD and ADAM optimizers. Crucially, the "Little is Enough" variant of the attack successfully circumvents the MultiKrum robust aggregation defense, which is a strong piece of evidence for its effectiveness.

The paper includes an analysis of how the attack's success is influenced by the constraints on the feasible set of poisons, which adds credibility to the results.

**Requested Changes:**

The paper would be significantly strengthened by a deeper discussion of the threat model's assumptions. Please add a section in the discussion or conclusion that explicitly addresses the practicality of an attacker having iterative access to model weights and the ability to inject data at each step.

Discuss potential ways the attack could be adapted for a more constrained, "one-shot" poisoning scenario.

To better position the work, I suggest evaluating the attack against at least one other class of defense, such as data sanitization methods (e.g., using outlier detection) or certified defenses. This would provide a more complete picture of the attack's robustness.

The references to "Result A" and "Result B" in the text (e.g., in Figure 1 and Section 4) are a bit confusing. I recommend making these labels more descriptive or integrating their descriptions more directly into the main text to improve readability.

The current title is functional but could be more impactful. A title like "Data Poisoning via Malicious Gradient Inversion" or "Equating Data Poisoning and Gradient Attacks via Gradient Inversion" might better capture the core contribution.

---

### Review · Reviewer_C5Be · 2025-08-26

**Summary Of Contributions:**

The paper argues that data poisoning can match the harm made by gradient attacks in non-convex settings. The authors provide positive answers through numerical experiments on (non-convex) neural networks by inverting malicious gradients into feasible training examples at each iteration for three gradient attacks: Gradient Ascent, Orthogonal Gradient, and Little is Enough.

**Audience:**

Yes

**Audience Explanation:**

This paper bridges the gap between data poisoning and gradient attacks in non-convex settings to some extent.

**Broader Impact Concerns:**

No Concern.

**Claims And Evidence:**

Yes

**Claims Explanation:**

The authors provide empirical evidence through numerical experiments.

**Requested Changes:**

(1) From the introduction, I am not motivated by the idea of matching the harm of data poisoning to the harm of gradient attacks.
Why are we studying this, and what can this result bring?

(2) The attacker needs per-iteration poisoning with access to current weights, which is much stronger than standard data poisoning requirements. Can the authors provide realistic justifications? This is also not clear for the claim: "as low as 1%" since $\alpha$ is chosen per iteration.

---

### Review · Reviewer_SQoY · 2025-09-28

**Summary Of Contributions:**

This paper investigates the effectiveness of privacy attacks, specifically gradient-based attacks and data poisoning. To facilitate comparison, the authors introduce a novel attack model and demonstrate that data poisoning attacks can replicate the behavior of gradient-based counterparts. These findings point to new research directions concerning the robustness of machine learning algorithms.

**Audience:**

Yes

**Audience Explanation:**

This work is relevant for ML research, especially in the areas of robustness and privacy.

**Claims And Evidence:**

Yes

**Claims Explanation:**

The Authors present carefully designed experiments that align with their proposed attack model. While certain assumptions in the attack model may be somewhat unrealistic, the results remain valid and valuable as a proof of concept.

**Requested Changes:**

The assumptions underlying the attack model appear somewhat impractical. It would be valuable to investigate the extent to which these assumptions influence the attack’s effectiveness.

- Analyzing how changes in the distribution of the auxiliary dataset (when diverging from the training data) impact attack success would provide further insights.

- Visualizing the poisoned data generated through gradient-based modifications could help assess their realism. For instance, do these examples appear plausible, or could such attacks be mitigated simply by removing outliers (assuming the distribution of poisoned data differs from that of the training data)?

---

### Review · Reviewer_rVzR · 2025-10-10

**Summary Of Contributions:**

This paper shows that data poisoning can replicate the harm of gradient attacks via gradient inversion in non-convex neural networks. The authors propose a threat model in which the attacker can inject poisoned samples at every training iteration, and they empirically demonstrate the effectiveness of data poisoning under different gradient attack strategies. Although the setup is idealized (assuming the attacker knows the model and weights at each iteration), it shows that data poisoning can achieve results comparable to gradient attacks in the worst-case scenario.

**Audience:**

Yes

**Audience Explanation:**

The paper studies whether data poisoning can achieve effects similar to gradient attacks in non-convex neural network settings. Its findings on gradient inversion and worst-case attacks will be of interest to researchers studying model robustness, security, and privacy.

**Broader Impact Concerns:**

No concern.

**Claims And Evidence:**

Yes

**Claims Explanation:**

The authors provide thorough empirical results through carefully designed experiments on CIFAR10 using both CNN and ViT-tiny. The results show that data poisoning crafted via gradient inversion can degrade model performance. They demonstrate that such attacks can achieve effects comparable to gradient attacks, supporting the claims made in the paper.

**Requested Changes:**

The motivation could be a bit clearer. It would help if the authors explained why showing that data poisoning can match gradient attacks matters in practice. For instance, the author could discuss real-world implications, like how understanding the limits of data poisoning can help evaluate AI system security or guide the design of more robust models. Making these connections clearer would show the study’s relevance and broader impact.

---

> ### Author Response · Authors · 2025-10-12
>
> We thank the reviewer for their positive feedback.\
> We appreciate your suggestion and made changes to the introduction to explicitly link the better understanding of attacks potential with the development of comprehensive security measures.
> We wanted this submission to remain within the 12 pages limit suggested in TMLR guidelines to be mindful of reviewers' timeline. A reference to Appendix A (which discuss real-world implications of data poisoning) has been added to the introduction.
> We expect this modification to address your concern and would be glad to discuss further otherwise.
>
> Thanks again for your time and help.

---

### Decision · Action_Editor_tvhm · 2025-11-06

**Recommendation:** Accept as is

**Additional Comments:**

The recommendation is based on the reviewers' comments, the action editor's evaluation, and the authors’ response.

This paper proposes advanced data poisoning attacks. While reviewers found the assumption of the considered threat model could be too strong to be realistic, they also found the studied setting novel, and the results provide new insights. The authors’ rebuttal has successfully addressed the major concerns of reviewers. Therefore, I recommend acceptance of this submission. I also expect the authors to include the new results and suggested changes during the rebuttal phase to the final version.

**Audience:**

Yes

**Audience Explanation:**

of broad interest to TMLR community

**Claims And Evidence:**

Yes

**Claims Explanation:**

The claims of advanced data poisoning attacks (under the assumptions of the considered threat model) are supported by the empirical results.